# Porcine Circovirus 3a Field Strains in Free-Living Wild Boars in Paraná State, Brazil

**DOI:** 10.3390/ani11061634

**Published:** 2021-05-31

**Authors:** Tatiana Carolina Gomes Dutra de Souza, Danielle Gava, Rejane Schaefer, Raquel Arruda Leme, Gisele da Silva Porto, Amauri Alcindo Alfieri

**Affiliations:** 1Laboratory of Animal Virology, Department of Veterinary Preventive Medicine, State University of Londrina, Rodovia Celso Garcia Cid-Campos Universitário, Londrina 86057-970, Brazil; tatianacdutra@hotmail.com (T.C.G.D.d.S.); raquelarrudaleme@gmail.com (R.A.L.); giselesilvaporto@gmail.com (G.d.S.P.); 2Embrapa Suínos e Aves, Concórdia 89715-899, Brazil; danielle.gava@embrapa.br (D.G.); rejane.schaefer@embrapa.br (R.S.)

**Keywords:** *Sus scrofa*, feral pig, circoviruses

## Abstract

**Simple Summary:**

Porcine circovirus 3 (PCV-3) was first identified in pigs in the USA and was subsequently detected in several other countries, including Brazil. PCV-3 can be associated with diseases in pigs. To date, there are only a few reports of PCV-3 in wild boars worldwide. This study aimed to investigate the presence of PCV-3 in wild boars in Paraná state, Brazil. The results revealed that PCV-3 was present in the serum and lungs of the sampled boars. The complete genome of the PCV-3a strain was determined and compared with other PCV-3 strains around the world. Phylogenetic analysis has shown a close relationship to the strains already described in domestic and wild pigs. At this moment, there is no evidence that PCV-3 causes disease in wild boars. However, the monitoring of circulation of PCV-3 in wild boars is important for pig industry biosecurity because these animals share pathogens with domestic pigs.

**Abstract:**

Porcine circovirus 3 (PCV-3) was identified in domestic pigs worldwide. Although PCV-3 has also been detected in wild boars, information regarding its circulation in this free-living animal species is scarce. To investigate PCV-3 occurrence in free-living wild boars in Brazil, 70 serum samples collected between January 2017 and June 2019 in Paraná state, Brazil were analyzed by PCR assay. Amplicons measuring 330 bp in length were amplified in seven (10.0%) of the serum samples and confirmed to be PCV3-specific by nucleotide (nt) sequencing. As the amplified products from the serum samples yielded only intermediate levels of viral DNA, lung samples from the seven PCR-positive wild boars were also evaluated by PCR. Of these samples, five lung samples were positive and provided high levels of viral DNA. The three lung samples that presented the highest levels of viral DNA were selected for amplification and sequencing of the whole PCV-3 genome. The three full-length sequences obtained were grouped in PCV-3 clade “a”, and the sequences exhibited 100% nucleotide similarity among them. The PCV-3 field strains of this study showed nucleotide and amino acid similarities of 98.5–99.8% and 98.8–100%, respectively, with whole-genome PCV-3 sequences from around the world.

## 1. Introduction

Porcine circovirus 3 (PCV-3) belongs to the genus *Circovirus* and was recently identified in the USA through metagenomic analysis [1,2]. Subsequently, PCV-3 has been reported in several countries of South America, Europe, and Asia [3,4,5,6,7,8,9,10]. Retrospective studies revealed PCV-3 circulation since 1993 in Sweden [11], 1996 in China [12] and Spain [5], and 2006 in Brazil [13].

The PCV-3 genome consists of 1999–2001 nucleotides (nt) of circular, single-stranded DNA featuring two major open reading frames (ORFs). ORF1 encodes the replicase protein (Rep), which is composed of 296–297 amino acids (aa); this ORF is the most conserved region of the genome and shares 55% aa identity with the Rep of porcine circovirus 2 (PCV-2) [1]. ORF2 is located on the negative strand and encodes the capsid protein (Cap), the only constituent of the viral capsid; it is composed of 214 aa, sharing approximately 26–37% identity with the PCV-2 Cap protein [1,2]. ORF3 encodes a putative 231-aa protein, and its function has not been elucidated [1].

The PCV-3 sequences available in GenBank have high nucleotide identity between strains [14]. The evolutionary analysis in phylogenetic studies indicate the presence of a common ancestor dated approximately 1967 [15]. Considering a maximum genetic distance of 3% within the complete genome and a bootstrap support higher than 90%, Franzo and colleagues [16] suggest only two clades that can be defined as genotypes. Specifically, PCV-3a forms clade 1, while PCV-3b form clade 2. To date, only two strains are included in clade 2 (GenBank access numbers MG372488 and MG372490).

PCV-3 has been detected in symptomatic [1,2] and asymptomatic pigs [14] and in other animal species, including dogs [17], cattle [18] and wild hosts [19]. The PCV-3 prevalence in domestic pigs varies from 6.5 to 68.6% [4,5,6,10,12,14], while in wild boars, it varies from 9.1 to 57.1% and which, due to their habits, may contribute to the spread of the virus [13,20,21,22]. PCV-3 infection in wild boars has been reported in Germany [20], Italy [21], Spain [22], and recently, in Brazil [13].

Brazil is the fourth-largest pork producer and exporter worldwide, and the state of Paraná is the second-largest pork producer in Brazil, accounting for 19.8% of total pork meat production in 2019 [23]. The total population of free-living wild boars in Brazil is unknown, but sightings are common in crop fields and near livestock farms in various Brazilian regions, including Campos Gerais, Paraná state [24]. This study attempted to investigate PCV-3 occurrence in free-living wild boars in Campos Gerais, Paraná state, and to genetically characterize the PCV-3 strains detected in this animal species.

## 2. Materials and Methods

### 2.1. Sample Collection

From January 2017 to January 2019, 70 free-living wild boars were harvested in the Campos Gerais region of the state of Paraná; specifically, 14 juvenile females, 14 juvenile males, 31 adult females, and 11 adult males were obtained. The classification of the animals into juvenile and adult animals was performed according to Hebeisen et al. (2008) [25].

Hunting was performed by exotic wildlife controller agents who were authorized by the Brazilian Institute of Environment and Renewable Natural Resources (IBAMA) according to IN 03/2013 [26], registered in the Federal Technical Register of Potentially Pollutive Activity (CTF/APP), and closely monitored by the Brazilian Army. Paired serum and lung samples were collected from all 70 free-living wild boars and stored at −80 °C. Sera were used to assess the frequency of PCV-3 infection in the studied population, as is commonly carried out in pioneering studies in wild boars [21,22].

After screening of serum samples, the lungs of the animals that had positive serum samples were evaluated for the presence of PCV-3 following the same methodology. This tissue was selected because it is a replication site of the virus and a high viral load can be found [22].

### 2.2. DNA Extraction and PCR

Viral DNA was extracted from serum samples using a DNeasy Blood & Tissue Kit (Qiagen, Valencia, CA, USA) according to the manufacturer’s instructions. PCR assays for PCV-3 diagnosis were performed using a pair of primers (5′-CCA CAG AAG GCG CTA TGT C-3′ and 5′-CCG CAT AAG GGT CGT CTT G-3′) that amplify a 330-bp fragment of the capsid gene [1] in 25-μL final reaction volume. The amplified products were analyzed by electrophoresis on a 1% agarose-TBE gel and stained with ethidium bromide.

### 2.3. Genome Sequencing

Complete genome sequencing of three PCV-3 PCR-positive lung samples was performed using primers 5′-CAC CGT GTG AGT GGA TAT AC C-3′, 5′-GTC GTC TTG GAG CCA AGT G-3′, 5′-TGT TGT ACC GGA GGA GTG-3′, and 5′-GAA GTT GCG GAG AAG ATG-3′, described by Palinski et al. (2017) [1] and a primer with degenerate 3′ end GCCGAC-TAATGCGTAGTCNNNNNNNNN described by Franzo et al. (2018) [6]. The selection of the three PCV-3 samples for sequencing was based on viral load and on wild boar geographic location.

The amplicons were purified using the PureLink^®^ Quick Gel Extraction Kit (Invitrogen, Carlsbad, CA, USA), quantified with a Qubit™ Fluorometer (Invitrogen™ Life Technologies, Eugene, OR, USA), and analyzed by electrophoresis on a 2% agarose gel. The ABI3500 Genetic Analyzer and BigDye™ Terminator v3.1 A Cycle Sequencing Kit (Applied Biosystems, Foster City, CA, USA) was used for sequencing, which was performed in both directions with the forward and reverse primers employed in the PCR assay. Quality assessment and sequence analyses were determined by Phred [27] and Lucy [28] software. The sequences were assembled using Cap3 [27] to generate consensus sequences.

### 2.4. Genome Analyses

Complete PCV-3 genome sequences were accessed on GenBank. The misaligned strings were excluded from the alignment. Therefore, only one sequence was selected as representative of all identical sequences. The construction of the tree comprised complete PCV-3 sequences from wild boars and domestic pigs from different continents and recent and contemporary strains. As reported by Franzo and colleagues [16], two representative sequences of the PCV-3b clade were also included.

Phylogenetic analysis of the complete genome sequences of PCV-3 was performed using the neighbor-joining (NJ) method in MEGA 6.0 software [29]. Bootstrap values were determined with 1000 replicates, and the evolutionary distances were computed using the Tamura 3 parameter model [29]. The genome sequences were compared with other PCV-3 sequences that are available in GenBank.

The complete PCV-3 sequences of this study and PCV-3 sequences from wild boars and domestic pigs from different continents and recent and contemporary strains were analyzed using BioEdit version 7.2.6.1 [30].

## 3. Results

PCV-3 was detected in 7 (10%) out of the 70 serum samples. PCV-3-positive samples were obtained from adult female boars from Castro and Ponta Grossa Counties. Additionally, five (71%) of the seven lung samples from wild boars with PCV-3-positive serum samples were positive. The complete genomes of the three sequenced PCV-3 Brazilian strains were deposited in GenBank (accession numbers: MT075517, MT075518, and MT075519). The sequences presented 100% nt similarity among themselves and were classified as PCV-3a (Figure 1).

The Brazilian PCV-3 wild boar strains showed 99.6% nt similarity with Chilean (MN907812) and Chinese strains (MG897494) and 99.5% similarity with American (KX966193) and Taiwanese strains (MK343155) identified from domestic pigs.

The PCV-3 sequences described in this study (MT075517, MT075518, and MT075519) revealed 98.7–99.8% nt and 99.0–100% aa similarity with the Brazilian strains from domestic pigs (MF079254, MF079253, MK645715, MK645718, MK645719, MK645716, and MK645717) (Table 1), 98.5–99.8% nt and 98.8–100% amino acid (aa) sequence similarity with PCV-3 sequences from domestic pigs worldwide, and 98.6–99.2% nt and 98.9–99.5% aa sequence similarity with PCV-3 sequences from free-living wild boars (MH579736; MH579747; MG820624; MH699985).

## 4. Discussion

The PCV-3 phylogenetic tree (Figure 1) showed that the PCV-3 sequences from this study were more similar to Brazilian, American, Taiwanese, Chinese and Chilean strains obtained from domestic pigs than to other PCV-3 nt sequences obtained from wild boar and available in GenBank. This result suggests that PCV-3 may circulate between Brazilian domestic and feral pigs. In addition, the notable similarity between strains from Brazilian wild boars and those obtained in different countries and years suggests the high genetic stability of PCV-3 field strains.

In this study, PCV-3 DNA was detected in 7 (10%) out of the 70 serum samples, suggesting a systemic infection. Previous studies have shown that wild boars are a potential reservoir for PCV-3 infection in wild and domestic pigs [22]. Additionally, according to Klaumann et al. (2018) [31], PCV-3 could be detected over a long time, suggesting that wild boars may exhibit long-lasting infections. Other viruses belonging to the *Circovirus* genus, such as PCV-2, also produce persistent viremia in pigs [32]. PCV-3 was reported in serum from wild boars in some countries. Italy was the first country to report PCV-3 infection in wild boars, where 33% of serum samples collected from 2014 to 2015 were positive [21]. In Spain, 42.6% of serum samples collected from wild boars from 2004 to 2018 were positive for PCV-3, demonstrating that the virus has been circulating since 2004 [22]. A longitudinal study in Spain, which involved the capture and recapture of wild boars for over a year, detected PCV-3 in serum samples of 52.6% of the evaluated animals [22].

The prevalence of PCV-3 DNA in serum samples of free-living wild boars in Brazil was lower (10%; 7/70) than that observed in European studies [21,22]. PCV-3 studies in Brazil are scarce but showed that PCV-3 has been present in domestic pigs since 2006 [4], and in wild boars since 2013 [13]. In the country, PCV-3 was previously detected in serum of sows presenting stillbirths [3] and the prevalence of PCV-3 in serum samples of domestic pigs was higher (26.7%; 41/154) [9] than observed in wild boars in our research. In a retrospective study, PCV-3 was detected in 47.8% (32/67) of different domestic pig samples such as lung, lymph nodes, and spleen; however, the prevalence for each organ was not reported [4]. A recent study by Varela et al. (2020) [13] detected PCV-3 in 36.3% (29/80) of retropharyngeal lymph nodes from wild boars captured from 2013 to 2015 in Rio Grande do Sul state; other samples types and animals age and sex were not evaluated.

Our study was the first to investigate the occurrence of PCV-3 in free-living wild boars in a specific Brazilian region (Paraná state, South Brazil), and to describe PCV-3 recovered from the serum and lungs of these species in Brazil. Selection of sample types for PCV-3 detection was based on previous studies that indicated spleen [20,22], tonsil [13,22], liver [22] and lungs [22] to be the most useful tissues for PCV-3 detection. Furthermore, lungs had the highest prevalence of positivity (57.1%) compared to other tissues [22], and they are considered as a target for PCV-3 replication [31]. Although the viral load in the submandibular lymph node was higher, the percentual of positivity in this tissue was lower compared to other tissues [22]. Serum is considered the most appropriate sample for epidemiological studies in wild boars, despite PCV-3 viral load being lower when compared with viral load found in other tissues [22]. So far, there are few published studies of PCV-3 detection in wild boars [13,20,21,22], and 66.6% of them use serum as biological material for initial PCV-3 investigation [21,22].

Studies published to date observed a high prevalence of PCV-3 in adult wild boars (47.5%) and a low prevalence in juveniles (8.6%) [22]. Surprisingly, in our study, PCV-3 was only detected in adult free-living wild boars, suggesting less virus circulation in juvenile free-living wild boars. Usually, adult males live alone while adult females are frequently seen in flocks with or without offspring. The lower PCV-3 detection observed in juvenile wild boars suggests a low viremia at the time of sample collection [10], possibly due to the presence of PCV-3 antibodies from maternal colostrum.

## 5. Conclusions

This report is the first to describe PCV-3a in free-living wild boars in Paraná state, South Brazil. The identity matrix demonstrated a high nt similarity among the three PCV-3 strains from this study and other strain sequences that are available in GenBank. PCV-3 prevalence in wild boars should be evaluated to determine the dynamics of virus evolution in this pig population.

## Figures and Tables

**Figure 1 animals-11-01634-f001:**
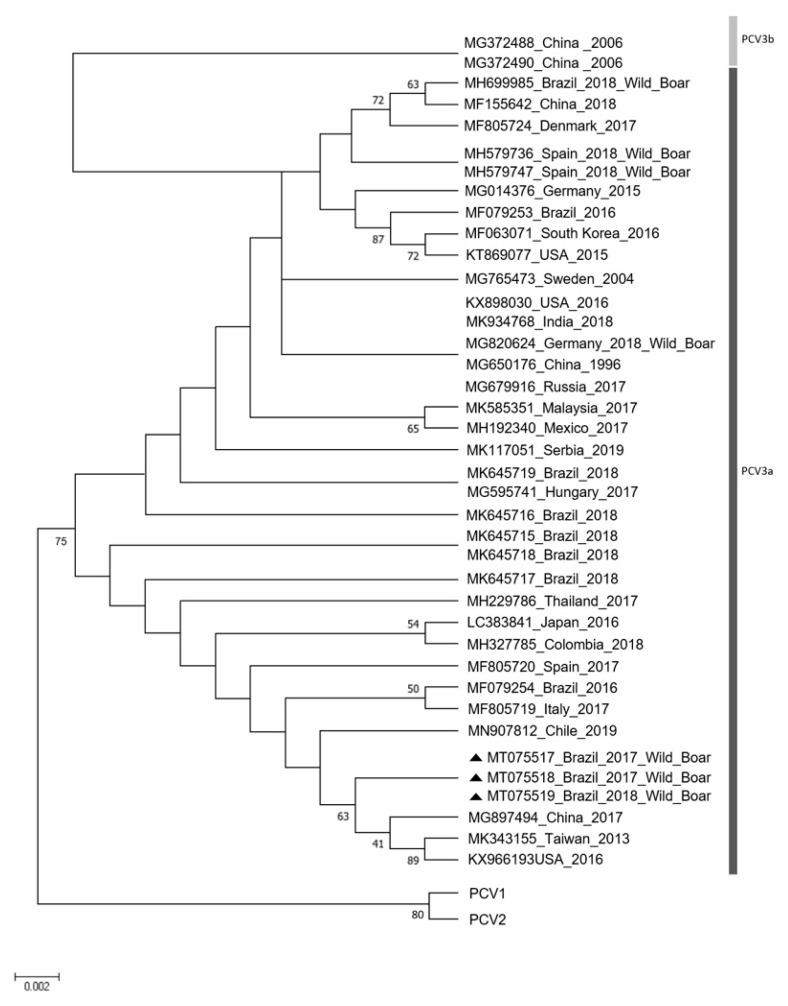
Phylogenetic tree of the full-length PCV-3 genome using the neighbor-joining method. Bootstrap values were determined with 1000 replicates, and the evolutionary distances were computed using the Tamura 3 parameter model. Evolutionary analyses were conducted with MEGA 6.0 software [29]. The three Brazilian PCV-3 field strains from free-living wild boars are labeled with a solid black triangle. The sequences of PCV-1 and PCV-2 were used as outgroups.

**Table 1 animals-11-01634-t001:** Similarities of nucleotides (nt) and amino acids (aa) of the complete genome of porcine circovirus 3 (PCV-3) sequences reported in this study and representative strains from other countries. The complete PCV-3 sequences were analyzed using BioEdit version 7.2.6.1 [30].

Representative PCV-3 Strains					Representative Strains in This Study					
Species	Continent	Country	Strain	Year	MT075517		MT075518		MT075519	
					nt	aa	nt	aa	nt	aa
Swine	Asia	China	MG650176	1996	99.6	99.8	99.6	99.8	99.6	99.8
		China	MG372488	2006	99.3	99.5	99.3	99.5	99.3	99.5
		China	MG372490	2006	99.3	99.5	99.3	99.5	99.3	99.5
		China	MG897494	2017	99.6	99.8	99.6	99.8	99.6	99.8
		China	MF155642	2018	98.9	99.1	98.9	99.1	98.9	99.1
		Taiwan	MK343155	2013	99.5	99.6	99.5	99.6	99.5	99.6
		Korea	MF063071	2016	98.7	99.0	98.7	99.0	98.7	99.0
		Japan	LC383841	2016	99.3	99.6	99.3	99.6	99.3	99.6
		Thailand	MH229786	2017	99.3	99.5	99.3	99.5	99.3	99.5
		Malaysia	MK585351	2017	99.5	99.8	99.5	99.8	99.5	99.8
		Russia	MG679916	2017	99.7	99.8	99.7	99.8	99.7	99.8
		India	MK934768	2018	98.6	98.8	98.6	98.8	98.6	98.8
	Europe	Denmark	MF805724	2017	98.9	99.1	98.9	99.1	98.9	99.1
		Germany	MG014376	2015	99.3	99.5	99.3	99.5	99.3	99.5
		Sweden	MG765473	2004	98.6	98.8	98.6	98.8	98.6	98.8
		Spain	MF805720	2017	99.6	99.8	99.6	99.8	99.6	99.8
		Hungary	MG597441	2017	99.7	99.8	99.7	99.8	99.7	99.8
		Italy	MF805719	2017	99.6	99.8	99.6	99.8	99.6	99.8
		Serbia	MK117051	2019	99.7	99.9	99.7	99.9	99.7	99.9
	North America	USA	KT869077	2015	98.7	99.1	98.7	99.1	98.7	99.1
		USA	KX898030	2016	98.5	98.8	98.5	98.8	98.5	98.8
		USA	KX966193	2016	99.5	99.6	99.5	99.6	99.5	99.6
		Mexico	MH192340	2017	99.5	99.8	99.5	99.8	99.5	99.8
	South America	Colombia	MH327785	2018	99.3	99.5	99.3	99.5	99.3	99.5
		Chile	MN907812	2019	99.6	99.8	99.6	99.8	99.6	99.8
		Brazil	MF079254	2016	99.6	99.9	99.6	99.9	99.6	99.9
		Brazil	MF079253	2016	98.7	99.0	98.7	99.0	98.7	99.0
		Brazil	MK645715	2018	99.6	100	99.6	100	99.6	100
		Brazil	MK645718	2018	99.8	100	99.8	100	99.8	100
		Brazil	MK645719	2018	99.8	100	99.8	100	99.8	100
		Brazil	MK645716	2018	99.6	100	99.6	100	99.6	100
		Brazil	MK645717	2018	99.3	99.8	99.3	99.8	99.3	99.8
Wild boar	Europe	Spain	MH579736	2005	99.2	99.5	99.2	99.5	99.2	99.5
		Spain	MH579747	2018	98.7	99.3	98.7	99.3	98.7	99.3
		Germany	MK280624	2018	98.6	99.1	98.6	99.1	98.6	99.1
	South America	Brazil	MH699985	2018	98.7	98.9	98.7	98.9	98.7	98.9

## Data Availability

The data that support the findings of this study are available on request from the corresponding author.

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
