# Peer review of "Porcine Circovirus 3a Field Strains in Free-Living Wild Boars in Paraná State, Brazil"

_animals, 2021, doi:10.3390/ani11061634_

Round 1
Reviewer 1 Report
In this study the authors aimed to investigate PCV-3 occurrence in free-living wild boars in Paraná state, Brazil and to genetically characterize the PCV-3 strains detected.
The article presents interesting data relating to a geographical area. The introduction is well written but could be improved.
Line 14: replace “US” with “USA”.
Lines 48-49: the sentence should be reworded. ORF2 is not a protein; ORF2 is located on the negative strand and encodes the Cap protein, the only constituent of the viral capsid, which is considered the most variable and most immunogenic viral protein.
The methods could be improved. From the methods and the abstract it is clear that the serum samples were analyzed first and then the lungs from the animals that had the positive serum sample. The authors should clarify why this choice was made and why not all the lung samples were analyzed since this organ is considered as a target for virus replication.
The results are clearly expressed but at least one aspect needs to be clarified: the authors reported that in 7 out of 70 serum samples PCV-3 was detected, 5 out of 7 lung samples tested positive for PCV-3; why only three PCV-3 were sequenced and from which matrix?
In the discussion section (lines 192-193) the sentences “Although serum is not known as the main target tissue or cell tropism of PCV-3, serum appears to be the most appropriate biological sample for detecting the virus” should be argued.
Author Response
Reviewer #1:
Point 1: In this study the authors aimed to investigate PCV-3 occurrence in free-living wild boars in Paraná state, Brazil and to genetically characterize the PCV-3 strains detected. The article presents interesting data relating to a geographical area. The introduction is well written but could be improved.
Response 1 : Thank you for your comment and suggestion. We agreed and restructured the introduction. The following sentences were added in the manuscript.
Lines 43 to 45, page 1: Retrospective studies revealed PCV-3 circulation since 1993 in Sweden [11], 1996 in China [12] and Spain [5], and 2006 in Brazil [13].
Lines 54 to 56, page 2: The PCV-3 sequences available in GenBank have high nucleotide identity between strains [15]. The evolutionary analysis phylogenetic studies indicate the presence of a common ancestor dated approximately 1967 [16].
Lines 63 to 66, page 2: The PCV-3 prevalence in domestic pigs varies from 6.5-68.6% [4–6,10,12,15], while in wild boars varies from 9.1-57.1% and which, due to their habits, may contribute to the spread of the virus [13,20–22]
Point 2:Line 14: replace “US” with “USA”.
Response 2: The initials were replaced. Please, see line 42
Point 3: Lines 48-49: the sentence should be reworded. ORF2 is not a protein; ORF2 is located on the negative strand and encodes the Cap protein, the only constituent of the viral capsid, which is considered the most variable and most immunogenic viral protein.
Response 3: We appreciate your comment. The sentence was reformulated. Please, see lines 50-52.
Lines 50-52: ORF2 is located on the negative strand and encodes the capsid protein (Cap), the only constituent of the viral capsid; it is composed of 214 aa, sharing approximately 26-37% identity with the PCV-2 Cap protein [1,2].
Point 4: The methods could be improved. From the methods and the abstract it is clear that the serum samples were analyzed first and then the lungs from the animals that had the positive serum sample. The authors should clarify why this choice was made and why not all the lung samples were analyzed since this organ is considered as a target for virus replication.
Response 4: Thank you for your comment, we agree with your suggestion and have included the explanation in the Materials and Methods section.
Lines 93-94: Sera were used to assess the frequency of PCV-3 infection in the studied population, as is commonly done in pioneering studies in wild boars [21,22].
Lines 95-98: After screening of serum samples, the lungs of the animals that had positive serum samples were evaluated for the presence of PCV-3 following the same methodology. This tissue was selected because it is a replication site of the virus and a high viral load can be found [22].
Point 5: The results are clearly expressed but at least one aspect needs to be clarified: the authors reported that in 7 out of 70 serum samples PCV-3 was detected, 5 out of 7 lung samples tested positive for PCV-3; why only three PCV-3 were sequenced and from which matrix?
Response 5: Thank you for your comment. From the five PCR-positive lungs, only three presented high levels of viral DNA, being selected for amplification and sequencing of the whole PCV-3 genome. This information was included in lines 115-116 as followed.
Lines 115-116: The selection of the three PCV-3 samples for sequencing was based on viral load and on wild boar geographic location.
Point 6: In the discussion section (lines 192-193) the sentences “Although serum is not known as the main target tissue or cell tropism of PCV-3, serum appears to be the most appropriate biological sample for detecting the virus” should be argued.
Response 6: Thank you for your suggestion, we reformulated the sentence. .
Lines 214-218: Serum is considered the most appropriate sample for epidemiological studies in wild boars, despite PCV-3 viral load is lower when compared with viral load found in other tissues [22]. So far there are few published studies of PCV-3 detection in wild boars [13,20–22], and 66.6% of them use serum as biological material for initial PCV-3 investigation [21,22].

Reviewer 2 Report
This study aimed to investigate the presence of PCV-3 in wild boars in Paraná state, Brazil. The results revealed that PCV-3 is present in serum and lungs of the sampled boars. The complete genome of PCV-3a strain was determined and compared with others PCV-3 strains around the world. Phylogenetic analysis suggested a close relationship to the strains already described in domestic and wild pigs. It is helpful to accumulate the data of PCV-3 in wild boars in Paraná state, Brazil, and needs a moderate revision.
Major comments:
- I am concerned the novelty of this study. In reference 17 cited in the manuscript, those authors have concluded that in Brazil, wild boars are susceptible to PCV3 infection, and a high detection ratio (42.6%) was reported in wild boar serum samples from 2004 to 2018. It is suggested that the new findings in this study different from the conclusion obtained in reference 17 should be detailed.
- Lymph nodes are preferred to be employed to detect PCVs in domestic pigs. In this study, paired serum and lung samples were collected from all 70 free-living wild boars, lymph nodes were not involved in this study. Authors are encouraged to discuss about this concern.
Minor comments:
- In “4. Discussion”, the last sentence “suggesting less virus circulation in free-living wild boars” should be “suggesting less virus circulation in juvenile free-living wild boars”.
Author Response
Reviewer #2:
This study aimed to investigate the presence of PCV-3 in wild boars in Paraná state, Brazil. The results revealed that PCV-3 is present in serum and lungs of the sampled boars. The complete genome of PCV-3a strain was determined and compared with others PCV-3 strains around the world. Phylogenetic analysis suggested a close relationship to the strains already described in domestic and wild pigs. It is helpful to accumulate the data of PCV-3 in wild boars in Paraná state, Brazil, and needs a moderate revision.
Point 1: I am concerned the novelty of this study. In reference 17 cited in the manuscript, those authors have concluded that in Brazil, wild boars are susceptible to PCV3 infection, and a high detection ratio (42.6%) was reported in wild boar serum samples from 2004 to 2018. It is suggested that the new findings in this study different from the conclusion obtained in reference 17 should be detailed.
Response 1: Thank you for your comment. We agree with the suggestion and reformulate the discussion for a better understanding. Reference “17”chanced to “13”. Please consider Varela et al as a reference 13.
Lines 194-202:
The prevalence of PCV-3 DNA in serum samples of free-living wild boars in Brazil was lower (10%; 7/70) than that observed in European studies [21,22]. PCV-3 studies in Brazil are scarce but showed that PCV-3 is present in domestic pigs since 2006 [4], and in wild boars since 2013 [13]. In the country, PCV-3 was previously detected in serum of sows presenting stillbirths [3] and the prevalence of PCV-3 in serum samples of domestic pigs was higher (26.7%; 41/154) [9] than observed in wild boars in our research. In a retrospective study, PCV-3 was detected in 47.8% (32/67) of different domestic pig samples such as lung, lymph nodes, and spleen; however, the prevalence for each organ was not reported [4].
Lines 202-205: A recent study by Varela et al., (2020) [13] detected PCV-3 in 36.3% (29/80) of retropharyngeal lymph nodes from wild boars captured from 2013 to 2015 in Rio Grande do Sul state; and other samples types and animals age and sex were not evaluated.
Lines 206-208: Our study was the first to investigate the occurrence of PCV-3 in free-living wild boars in a different Brazilian region (Paraná state, South Brazil), and to describe PCV-3 recovered from the serum and lungs of these species in Brazil.
Point 2: Lymph nodes are preferred to be employed to detect PCVs in domestic pigs. In this study, paired serum and lung samples were collected from all 70 free-living wild boars, lymph nodes were not involved in this study. Authors are encouraged to discuss about this concern.
Response 2: Thank you for your comment. We reformulated the paragraph to discuss better about tissue selection for PCV-3 detection. In addition, we have included the explanation in the Materials and Methods section.
Lines 208-214: Selection of sample types for PCV-3 detection was based on previous studies that indicated spleen [20,22], tonsil [13,22], liver [22] and lungs [22] as the most useful tissues for PCV-3 detection. Besides, lungs had the highest prevalence of positivity (57.1%) compared to other tissues [22], and they are considered as a target for PCV-3 replication [31]. Although the viral load in submandibular lymph node was higher, the percentual of positivity in this tissue was lower compared to other tissues [22].
Lines 93-94: Sera were used to assess the frequency of PCV-3 infection in the studied population, as is commonly done in pioneering studies in wild boars [21,22].
Lines 95-98: After screening of serum samples, the lungs of the animals that had positive serum samples were evaluated for the presence of PCV-3 following the same methodology. This tissue was selected because it is a replication site of the virus and a high viral load can be found [22].
Point 3: In “4. Discussion”, the last sentence “suggesting less virus circulation in free-living wild boars” should be “suggesting less virus circulation in juvenile free-living wild boars”.
Response 3: Thank you for your suggestion. We rephrased the sentence.
Lines 220-222: Surprisingly, in our study, PCV-3 was only detected in adult free-living wild boars, suggesting less virus circulation in juvenile free-living wild boars.

Round 2
Reviewer 2 Report
All my comments have been placed down, the paper can be accepted.